# Spatio-Temporal Landmark Detection via Selective Fine-Tuning of Echocardiography Foundation Models

Preetraj Bhoodoo[*1,2], Sarina Thomas[1,2,3], Elisabeth Wetzer[2,4], Anne Solberg[1,2], and Guy Ben-Yosef[5]

[1]Department of Informatics, University of Oslo, Oslo, Norway
[2]SFI Visual Intelligence, Tromsø, Norway
[3]Department of Cardiovascular Ultrasound, GE HealthCare, Oslo, Norway
[4]Department of Physics and Technology, UiT The Arctic University of Norway, Tromsø, Norway
[5]GE HealthCare Technology & Innovation Center, Niskayuna, New York, USA
 preetrb@uio.no

## Abstract

Foundation models (FMs) have shown remarkable capabilities across computer vision tasks, yet their effectiveness for complex medical downstream tasks remains underexplored. This work investigates whether state–of–the–art video–based FMs for echocardiography can perform precise spatio–temporal landmark detection without extensive fine–tuning. We evaluate two recent powerful FMs, namely EchoPrime, and PanEcho, pre–trained on few millions of echocardiographic video–text pairs, for left–ventricular contour detection at end–diastole (ED) and end–systole (ES) on EchoNet–Dynamic. We compare encoder regimes (frozen, partially frozen, fully trainable) and decoder heads (multi-layer perceptron (MLP) vs. graph convolutional network (GCN)), and benchmark against strong non–FM backbones (ResNet–18 2D/3D, ViT–Base, MViTv2–Small). Frozen encoders perform poorly and variably ($\approx$78.00 Dice, ED), whereas selectively unfreezing two blocks with GCN+augmentation yields a large jump ($91.71 \pm 3.49$ Dice, ED), recovering most of the improvement. Fully trainable EchoPrime (GCN+augmentation) achieves $93.13 \pm 3.11/90.95 \pm 3.71$ Dice (ED/ES), which is SOTA for regression-based models on EchoNet. Deploying separate, fully fine–tuned models for each task quickly becomes impractical in resource–constrained settings. Our results suggest that partially fine-tuning the FM is a resource-efficient strategy that recovers most of the performance benefits of end-to-end training, while avoiding the overhead of maintaining a separate model for each task. The code is available at https://github.com/preetrajb/EchoVLMLandmarks.

## 1 Introduction

Foundation models (FMs) have reshaped artificial intelligence, showing strong generalization across diverse tasks and modalities through large-scale self-supervised pre-training [1, 2]. Typically based on transformer architectures and trained on vast unlabeled datasets, these models often achieve robust performance with minimal fine-tuning [3]. While linear probing has yielded strong results in classification tasks, including echocardiography [4], it remains unclear whether frozen FM encoders suffice for more complex medical imaging tasks. In such cases, fine-tuning specific layers or the full encoder may be necessary, raising concerns about efficiency, overfitting, and loss of generalizability. Selective layer unfreezing and lightweight decoder heads have been proposed as resource-efficient alternatives [5], but their effectiveness for dense, spatio-temporal tasks remains underexplored.

In echocardiography, several FMs have been proposed for classification, segmentation, report generation, and diagnostic regression [4, 6–9]. Most are trained on large proprietary datasets, limiting opportunities for benchmarking on public data. One critical downstream task is the extraction of the left ventricular (LV) contour across the cardiac cycle, essential for measuring volumes and ejection fraction. Those are key indices for diagnosing and monitoring heart failure, cardiomyopathies, ischemic heart disease, and valvular heart disease. The EchoNet-Dynamic dataset [10] provides a benchmark for this task, with annotations at end-diastole (ED) and end-systole (ES). Prior approaches have relied on supervised segmentation [10, 11] or keypoint-based methods using heatmaps, regression, or spatio-temporal graphs [12, 13]. However, whether video-based FM encoders can be directly leveraged for spatio-temporal landmark prediction remains an open question, particularly since their architectures typically do not process frames independently.

Recent advances in self-supervised learning aim to address limited labeled data. EchoCLIP [7] introduced single-frame, single-view contrastive learning, while EchoPrime [4] extended this with large-scale video-based training on text–video pairs for reporting and measurement. PanEcho [8] scaled training to over one million videos for diagnostic prediction,

---

*Corresponding Author.

Proceedings of the 7th Northern Lights Deep Learning Conference (NLDL), PMLR 307, 2026.

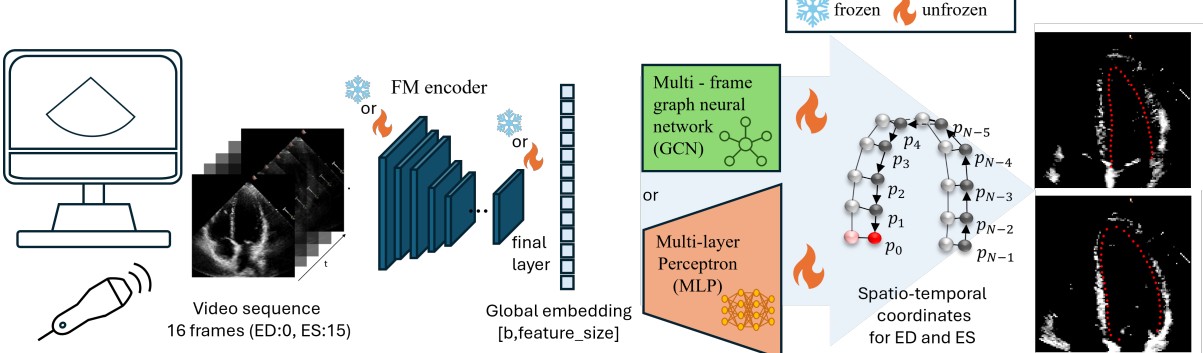

**Figure 1.** Overview of the experiment setup. An echocardiography video is sampled with 16 frames from end–diastole (ED) to end–systole (ES) and fed to a video-based FM encoder. The resulting embeddings serve as input for different heads types (multilayer perceptron (MLP) and graph convolutional network (GCN)) to reconstruct the spatio-temporal left ventricle contour in ED and ES.

and EchoFM [9] employed spatio-temporal masking and periodic-driven contrastive learning to capture cardiac dynamics. EchoApex [6] proposes a unified in–domain encoder with task–specific adapters trained on over 20M images for classification, measurement, and segmentation. Public weights for EchoApex are not available, which limits reproducibility on public benchmarks. Recent ultrasound foundation models such as USFM, URFM, Ultra-SAM, and FetalCLIP, while representing important advances in multi-organ imaging, interactive segmentation, and fetal ultrasound understanding, focus on different data domains and objectives than the spatio-temporal cardiac landmark detection task considered here [14–17].

Despite these advances, no prior work has applied video-based FMs to landmark detection, which requires both spatial precision and temporal consistency.

In this paper, we show that:

- State-of-the-art video-based foundation models can effectively perform multi-frame LV contour detection on EchoNet–Dynamic when paired with selective fine–tuning achieving performance competitive with strong non–FM baselines.

- Selective freezing strategies can substantially enhance temporal representation and yield accuracy close to full end-to-end training, while being more resource-efficient.

- Decoder head design impacts performance, with graph-based decoders providing consistent gains by better capturing spatial and temporal dependencies.

- FM-based models retain competitive performance even under reduced training data, demonstrating improved data efficiency compared to conventional architectures.

## 2    Methods

Video-based foundation models (FM) for echocardiography have demonstrated strong performance in classification and diagnostic regression tasks [4, 6–9]. In this paper, we investigate how these powerful models can be adapted for more complex, structured prediction tasks such as left-ventricular contour detection. We explore different transfer learning strategies, ranging from using the FM as a frozen feature extractor to partially or fully fine-tuning the encoder. This allows us to assess the trade-offs between performance and computational cost when applying a state-of-the-art echocardiography FM to a dense prediction task. In this section, we describe the chosen encoders, the downstream task formulation, and the decoder head designs used to bridge the adapted representations to landmark predictions.

### 2.1    Encoder design and pre-training

Our experiments focus on recent echocardiography foundation models, including the EchoPrime model [4], and the PanEcho model [8]. The EchoPrime encoder [4] is a vision transformer trained using vision-language contrastive learning on 12 million echocardiographic video-text pairs. Its underlying architecture is a Multiscale Vision Transformer (MViT-v2-S, [18]), comprising: (1) a 3D convolutional patch embedding layer; (2) learnable 3D positional encodings; and (3) a hierarchy of 16 multiscale transformer blocks. The PanEcho model [8] is another fully supervised, multitask deep learning model trained end-to-end on more than one million labeled echocardiography videos. Its architecture consists of a 2D convolutional image encoder that produces frame-level embeddings, followed by a temporal transformer that models sequential associations across frames.

Of particular interest in this work are 3D video-based encoders that produce a global video representation, such as the 512-dimensional embedding from

the MViT-v2 architecture used by EchoPrime [4]. A limitation of this design is progressive temporal compression, which reduces frame-level resolution in deeper layers. Nonetheless, prior work [13] suggests that such embeddings can still provide sufficient information for dense prediction tasks when coupled with appropriate decoders or fine-tuning strategies.

## 2.2 Landmark localization task:

Spatio-temporal landmark localization involves detecting a defined number $N$ of target points $P_t = \{\mathbf{p}_t^{(1)}, \mathbf{p}_t^{(2)}, \ldots, \mathbf{p}_t^{(N)}\}$ where $t$ denotes the time point and each landmark $\mathbf{p}_t^{(i)} \in \mathbb{R}^2$ is defined by two image pixel coordinates.

$$\mathbf{p}_t^{(i)} = [\mathbf{x}_t^{(i)}, \mathbf{y}_t^{(i)}], \quad i = 1, \ldots, N.$$

In most publicly available echocardiography datasets, only a subset of frames is annotated, typically at end-diastole (ED) and end-systole (ES). These two phases represent the largest and smallest left ventricular volumes during the cardiac cycle and are the standard reference points for calculating clinically important measures such as ejection fraction. As a result, intermediate frames remain unlabeled even though they contain valuable temporal context. To exploit this information, the video encoder processes all frames of the sequence, while supervision is applied only at annotated time points. The overall task design and encoder–decoder architecture are illustrated in Fig. 1.

## 2.3 Landmark decoder head design

The encoder produces a fixed global embedding that must be decoded back into spatial and temporal coordinates to predict landmarks on the annotated keyframes. We consider two decoder variants in this work.

**MLP decoder.** The first approach is a multilayer perceptron (MLP), where the pooled encoder features are passed through a stack of three fully connected layers with ReLU activations and dropout. The final layer outputs $N \times 2$ values, corresponding to the $(x, y)$ coordinates of each landmark. Despite its simplicity, this baseline often performs competitively on regression tasks and provides a reference point against which more structured decoders can be evaluated.

**GCN decoder.** Inspired by [13], we also employ a graph convolutional network (GCN) as the decoder head. In this design, the set of landmarks across time is modeled as a spatio-temporal graph, where each node corresponds to a keypoint at a specific frame. Edges are defined in two ways: (i) spatial edges connect adjacent keypoints within a contour, encoding local shape smoothness, and (ii) temporal edges connect the same landmark across neighboring frames, enforcing temporal coherence. Graph convolution layers propagate information along these edges, allowing the decoder to leverage both anatomical priors and motion dynamics. This design is particularly suited for cardiac structures such as the left ventricle, which maintains a closed and relatively stable contour across the cardiac cycle. To implement this, the graph is constructed using a cyclic adjacency for contour nodes together with explicit forward (and when applicable backward) temporal connections, which are processed through stacked SpiralConv layers to produce landmark coordinates. In the remainder of the paper, we denote these two decoder heads as **MLP** and **GCN**.

## 2.4 Baselines comparisons

We compare the performance of the FMs against a set of widely used backbone architectures, summarized in Table 1. These include both domain-specific supervised models and general-purpose backbones pretrained on large natural or action recognition datasets.

**EchoNet.** As a reference point from prior work, we include the supervised EchoNet-Dynamic model [10], which uses a 3D spatiotemporal CNN trained directly on echocardiography videos. This provides a domain-specific baseline trained end-to-end on over 10,000 labeled videos, representing the conventional supervised benchmark.

**ResNet-18 (2D).** As a convolutional baseline, we employ a 2D ResNet-18 initialized with ImageNet-1K weights. For video input, frames are processed independently and features are averaged before decoding, effectively providing an *early-fusion* representation across time. This model emphasizes strong spatial feature extraction but relies on temporal averaging for sequence information.

**ViT-Base (2D).** For transformer-based image encoding, we use a ViT-Base pretrained on ImageNet-21K. Patch embeddings are extracted per frame, with features from multiple frames averaged before decoding, analogous to the 2D ResNet baseline. This setup evaluates the transformer's capacity for spatial modeling with limited temporal aggregation.

**ResNet-18 (3D).** To capture spatio-temporal patterns explicitly, we evaluate a 3D ResNet-18 pretrained on Kinetics-400, which is a large-scale human action recognition dataset. This backbone processes short clips as spatio-temporal volumes, enabling comparison with 2D models that aggregate temporal context only at the feature level.

**MViT-v2-S.** Finally, we include the Multiscale Vision Transformer (MViT-v2-S), pretrained on Kinetics-400. This architecture, which combines hierarchical attention with multiscale temporal modeling, is structurally identical to the encoder used in EchoPrime [4]. Evaluating MViT-v2-S along-

side EchoPrime therefore enables a direct comparison of the effect of pretraining: action-recognition data (Kinetics) versus large-scale domain-specific vision–language pretraining (EchoPrime).

Together, these baselines cover both convolutional and transformer-based architectures, spanning natural image pretraining, large-scale action recognition, and domain-specific supervised learning. This allows us to assess how a video-based FM trained specifically on echocardiography (EchoPrime) compares against both established clinical baselines and strong general-purpose backbones.

# 3 Experiments & Results

## 3.1 Experimental Setup

To evaluate the effectiveness of frozen foundation model encoders for landmark detection tasks, we conduct comprehensive experiments using a large public echocardiography dataset and various baseline architectures.

### 3.1.1 Dataset and Data Preprocessing

We evaluate our approach on the EchoNet-Dynamic dataset [10], which contains 10,030 apical-4-chamber echocardiography videos acquired at Stanford University Hospital. Neither EchoPrime nor PanEcho was pretrained on EchoNet-Dynamic or its derivatives, and there is no overlap between their pretraining data and the test set. Each study includes labels for the end–diastolic (ED) and end–systolic (ES) frames (frame indices), clinical measurements (ejection fraction, ventricular volumes), and left–ventricular (LV) endocardial contours provided as 40 ordered keypoints at ED and ES. The original data are stored as AVI-format ultrasound videos. We extract clinically relevant frames and annotations by following the EchoNet-Dynamic protocol. For each case, segmentation masks and keypoints are generated from manual tracings provided in the dataset. ED and ES frames are determined by comparing LV cavity areas across all annotated frames: the frame with the maximum cavity corresponds to ED, and the minimum to ES. To capture temporal dynamics, we uniformly sample 16 frames spanning the interval between ED and ES within a cardiac cycle. Frames are resized to $224 \times 224$ pixels to match the EchoPrime input resolution. Following the official EchoNet-Dynamic setup, we use 7,460 videos for training, 1,277 for validation, and 1,288 for testing. To evaluate generalization in limited-data regimes, we further subsample the training set into smaller proportions: 0.5% (36 train / 6 val), 1% (73 / 12), 2% (147 / 25), 10% (739 / 127), 25% (1,848 / 318), 50% (3,697 / 636), and 100% (7,394 / 1,273).

### 3.1.2 Data Augmentation

To improve robustness to variability in ultrasound acquisition and imaging conditions, we employ a strong augmentation pipeline applied consistently across all frames in each video sequence, with keypoints transformed accordingly.

**Spatial transforms.** At the frame level, either a random resized crop (resizing back to the original size of $224 \times 224$) or a random shift–scale–rotate transform is applied with probability 0.6, introducing geometric variability.

**Color transforms.** With probability 0.6, one of several intensity-based operations is applied: random brightness/contrast adjustment, hue–saturation–value shift, or gamma correction.

**Blurring.** To simulate speckle and probe-related blur, Gaussian or median blur is applied with probability 0.3.

**Noise.** With probability 0.3, Gaussian or multiplicative noise is added to emulate acquisition noise.

By applying these transformations stochastically across training samples(same order as listed here), the model is encouraged to learn features invariant to common ultrasound artifacts while preserving anatomical consistency.

### 3.1.3 Encoder Freezing Strategies

All experiments were implemented in PyTorch, using the official torchvision release of MViT. Unless otherwise noted, models were trained up to 100 epochs using the Adam optimizer (batch size 20) on NVIDIA A100 GPUs. Input sequences consisted of 16 frames resized to $224 \times 224$ pixels. Training employed an initial learning rate of $5 \times 10^{-4}$, with a ReduceLROnPlateau scheduler that reduced the learning rate by a factor of 0.5 when validation loss plateaued for 5 consecutive epochs, enabling adaptive learning rate adjustment throughout training.

**Backbone structure.** Both EchoPrime and MViT-v2-S employ a Multiscale Vision Transformer backbone with 16 sequential transformer blocks. We denote by $k$ the number of blocks that are unfrozen during fine-tuning: $k = 0$ corresponds to a fully frozen backbone, $k = 2$ indicates that only the last two blocks are trainable, and $k = 16$ corresponds to full fine-tuning of all blocks. This setup allows us to systematically investigate the contribution of partial versus full encoder adaptation.

**Freezing regimes.** For the EchoPrime encoder, we evaluate three regimes under both augmentation and non-augmentation settings: (1) *Fully frozen* ($k = 0$): only the decoder head is trained; (2) *Partially unfrozen* ($k = 2$): the final two transformer blocks are trainable; (3) *Fully unfrozen* ($k = 16$): the entire encoder is fine-tuned.

**PanEcho encoder.** The PanEcho backbone consists of a 2D convolutional image encoder followed

**Table 1.** Overview of foundation models used in this study: Input (number of video frames), pre-training dataset, domain, and parameter counts. Note that $^\dagger$ indicates the same backbone architecture as the EchoPrime encoder.

| Model | Input | Pre-train data | Domain | Params |
|---|---|---|---|---|
| EchoNet [10] | 32 | 10,030 echo videos | Echo video | 39.6M |
| ResNet-18 (2D) [19] | 1 | ImageNet-1K [20] | Natural | 11.69M |
| ResNet-18 (3D) [21] | 16 | Kinetics-400 [22] | Action | 33.37M |
| ViT-Base (2D) [23] | 1 | ImageNet-21K [20] | Natural | 86.57M |
| MViT-v2-S$^\dagger$ [18] | 16 | Kinetics-400 [22] | Action | 34.5M |
| EchoPrime [4] | 16 | 12M echo-report pairs | Echo + text | 34.5M |
| PanEcho [8] | 16 | 1.23M echo videos | Echo video | ∼57M |

by a temporal transformer. In our experiments, we treat this spatial–temporal stack as a single backbone and consider two freezing configurations: a *fully frozen* setting ($k = 0$), where only the landmark decoder is trained on top of the pretrained PanEcho features, and a *fully trainable* setting ($k = 16$), where both the image encoder and temporal transformer are fine-tuned end-to-end using the same optimization hyperparameters and augmentation pipeline as for EchoPrime.

**Baselines.** For comparison, we train ResNet-18 (2D), ResNet-18 (3D), ViT-Base, and MViT-v2-S from their pretrained initialization (ImageNet or Kinetics) with full encoder fine-tuning. Each baseline is evaluated with both MLP and GCN decoder heads, with and without augmentation.

**Limited data experiments.** To test robustness in data-scarce settings, we repeat the freezing experiments (k = 0, 2, 16) using only 10% of the training data, comparing EchoPrime and MViT-v2-S with the GCN head.

**Scaling with data availability.** Finally, we explore the effect of training set size by performing full fine-tuning across data splits of 0.5%, 1%, 2%, 10%, 25%, 50%, and 100%. We additionally analyze how varying $k$ (frozen, partial, full) impacts performance across four representative splits (100%, 25%, 10%, and 2%).

This experimental design provides a comprehensive assessment of encoder adaptation strategies, comparing general-purpose and domain-specific pre-training across a range of data availability scenarios.

### 3.1.4 Evaluation metrics

To comprehensively assess landmark detection, we use two metrics. **Mean Keypoint Error (MKE)** computes the Euclidean distance between predicted and ground–truth landmarks, averaged across all keypoints for both keyframes (ED/ES). This provides a direct measure of spatial accuracy for landmark localization. **Dice score** is computed by converting predictions and ground truth to segmentation masks. To ensure comparability, both metrics are computed after mapping predictions back to the original EchoNet–Dynamic frame size of $112 \times 112$. For each test sample, Dice and MKE were computed

for ED and ES frames, and the mean and standard deviation of these metrics were reported to summarize both accuracy and consistency across the test set. This metric ensures compatibility with existing echocardiography benchmarks.

### 3.2 Results

We evaluate five backbones: ResNet–18 (2D), ResNet–18 (3D), EchoPrime, MViTv2–Small, and ViT–Base, on EchoNet–Dynamic (test set $n=1264$) using 16–frame, 224×224 inputs and report Dice and MKE at ED/ES. Overall, MViTv2–Small and EchoPrime attain the strongest accuracy (93.05 and 93.13 Dice at ED respectively), with 3D ResNet competitive and 2D ResNet/ViT trailing (Table 2). Our analysis is organized in two parts. Section 3.2.1 focuses on *non–foundation* backbones (ResNet–18 2D/3D, MViTv2–Small, ViT–Base), characterizing the impact of architecture, decoder head, and augmentation. Section 3.2.2 then analyzes *foundation models* (EchoPrime and PanEcho) under different encoder freezing regimes and compares their performance to the strongest non–foundation baseline.

### 3.2.1 Baseline Model Performance

Table 2 presents the performance of non-foundation model baselines on the EchoNet-Dynamic test set. The results establish several key findings about conventional architectures for landmark detection:

**3D vs. 2D processing.** Across heads and augmentation, ResNet–18 (3D) consistently surpasses its 2D counterpart (Table 2). With GCN heads, 3D attains 92.76/90.48 (ED/ES, with augmentation) and 92.83/90.41 (no augmentation) versus 91.42/88.79 and 91.63/89.02 for 2D gains of ≈+1.2–1.7 Dice. MKE is likewise lower by ≈0.25–0.40 px (e.g., ED: 2.31 vs. 2.71 with augmentation; ES: 2.19 vs. 2.52). Similar margins hold with MLP heads. *Takeaway:* explicit spatio–temporal modeling with 3D (video-based) encoding improves landmark accuracy and localization over sequential 2D processing.

**Transformer vs. Convolutional architectures.** Table 2 shows that a plain 2D ViT–Base is competitive but trails strong CNNs: with GCN+aug it attains 91.64/88.90 (ED/ES) and MKE 2.61/2.51,

**Table 2. Performance of different architectures** with/without augmentation and different heads: Keypoint localization and segmentation accuracy evaluated on EchoNet-Dynamic testset (n=1264) with annotated ED and ES frames (MKE = mean L1 keypoint pixel error). Dice scores are reported as percentages (mean ± standard deviation). A $112 \times 112$ grid is considered. EchoNet reports the original paper baseline dice results for single frames. All models use 16-frame input sequences, except the EchoNet baseline, which uses 32 frames. Best values are in **bold** and second-best are underlined.

| Architecture | Aug. | Head | Dice ED ↑ | Dice ES ↑ | MKE ED ↓ | MKE ES ↓ |
|---|---|---|---|---|---|---|
| EchoNet [10] | – | – | 92.78 | 90.68 | – | – |
| ResNet-18 (2D) | Yes | GCN | 91.42 ± 3.82 | 88.79 ± 4.90 | 2.71 ± 1.05 | 2.52 ± 1.02 |
| | Yes | MLP | 90.54 ± 4.20 | 87.71 ± 5.21 | 2.92 ± 1.07 | 2.77 ± 1.05 |
| | No | GCN | 91.63 ± 3.77 | 89.02 ± 4.87 | 2.65 ± 1.01 | 2.47 ± 0.99 |
| | No | MLP | 90.50 ± 4.05 | 87.89 ± 5.17 | 2.93 ± 1.08 | 2.72 ± 1.08 |
| ResNet-18 (3D) | Yes | GCN | 92.76 ± 3.28 | 90.48 ± 4.07 | 2.31 ± 0.95 | 2.19 ± 0.88 |
| | Yes | MLP | 91.66 ± 3.59 | 89.05 ± 4.43 | 2.62 ± 1.02 | 2.48 ± 0.93 |
| | No | GCN | 92.83 ± 3.21 | 90.41 ± 4.19 | 2.30 ±0.96 | 2.22 ± 0.95 |
| | No | MLP | 91.55 ± 3.54 | 88.75 ± 4.68 | 2.63 ± 1.01 | 2.53 ± 1.01 |
| ViT-Base (2D) | Yes | GCN | 91.64 ± 3.69 | 88.90 ± 4.97 | 2.61 ± 1.05 | 2.51 ± 1.07 |
| | Yes | MLP | 90.86 ± 3.87 | 87.98 ± 5.06 | 2.86 ± 1.08 | 2.72 ± 1.10 |
| | No | GCN | 89.18 ± 4.91 | 85.72 ± 6.44 | 3.29 ± 1.30 | 3.19 ± 1.32 |
| | No | MLP | 90.36 ± 4.10 | 87.07 ± 5.30 | 2.98 ± 1.13 | 2.88 ± 1.15 |
| MViTv2-Small | Yes | GCN | **93.05** ± 3.20 | **90.84** ± 3.93 | **2.26** ± 0.92 | **2.14** ± 0.85 |
| | Yes | MLP | 91.29 ± 3.80 | 88.40 ± 4.82 | 2.67 ± 1.03 | 2.58 ± 1.00 |
| | No | GCN | 92.75 ± 3.11 | 90.35 ± 4.16 | 2.34 ± 0.93 | 2.22 ± 0.89 |
| | No | MLP | 91.45 ± 3.59 | 88.60 ± 4.69 | 2.64 ± 1.02 | 2.56 ± 0.99 |

**Table 3. Performance under freezing/augmentation/head settings** for *EchoPrime* and *PanEcho* on EchoNet–Dynamic (test $n$=1264). Dice is reported as % (mean ± std) and MKE in pixels (mean ± std). A $112 \times 112$ grid is considered. For PanEcho, only fully frozen ($k$=0) and fully unfrozen ($k$=16) settings are included. All models use 16-frame input sequences. Best values per column are in **bold**, second–best are underlined.

| Backbone setting | Aug. | Head | Dice ED ↑ | Dice ES ↑ | MKE ED ↓ | MKE ES ↓ |
|---|---|---|---|---|---|---|
| **EchoPrime backbone** | | | | | | |
| Fully frozen | Yes | GCN | 78.41 ± 10.28 | 71.96 ± 13.85 | 6.81 ± 3.08 | 6.50 ± 3.09 |
| *(k = 0)* | Yes | MLP | 75.72 ± 10.85 | 68.62 ± 14.06 | 7.58 ± 3.43 | 7.19 ± 3.36 |
| | No | GCN | 79.80 ± 9.81 | 73.25 ± 13.48 | 6.31 ± 2.99 | 6.05 ± 3.01 |
| | No | MLP | 77.81 ± 10.40 | 71.16 ± 13.78 | 6.93 ± 3.32 | 6.61 ± 3.29 |
| Partially frozen | Yes | GCN | 91.71 ± 3.49 | 89.05 ± 4.61 | 2.63 ± 0.99 | 2.49 ± 0.97 |
| *(2 blocks unfrozen, k = 2)* | Yes | MLP | 90.82 ± 3.83 | 87.74 ± 5.08 | 2.85 ± 1.04 | 2.73 ± 1.06 |
| | No | GCN | 90.89 ± 3.99 | 88.05 ± 5.27 | 2.90 ± 1.16 | 2.71 ± 1.09 |
| | No | MLP | 91.40 ± 3.60 | 88.41 ± 4.83 | 2.70 ± 1.02 | 2.61 ± 1.03 |
| Fully trainable | Yes | GCN | **93.13** ± 3.11 | **90.95** ± 3.71 | **2.25** ± 0.92 | **2.11** ± 0.81 |
| *(k = 16)* | Yes | MLP | 91.32 ± 3.68 | 88.49 ± 4.75 | 2.67 ± 1.00 | 2.57 ± 1.00 |
| | No | GCN | 90.41 ± 4.19 | 88.35 ± 4.78 | 3.05 ± 1.14 | 2.72 ± 1.04 |
| | No | MLP | 91.74 ± 3.30 | 89.06 ± 4.41 | 2.61 ± 0.97 | 2.52 ± 0.95 |
| **PanEcho backbone** | | | | | | |
| Fully frozen | Yes | GCN | 78.33 ± 9.85 | 72.38 ± 13.25 | 7.01 ± 3.04 | 6.52 ± 3.04 |
| *(k = 0)* | Yes | MLP | 78.34 ± 10.04 | 72.31 ± 13.43 | 6.94 ± 3.20 | 6.47 ± 3.20 |
| | No | GCN | 77.25 ± 9.85 | 71.51 ± 13.15 | 7.61 ± 3.06 | 6.82 ± 3.13 |
| | No | MLP | 79.45 ± 9.72 | 73.41 ± 13.03 | 6.50 ± 2.98 | 6.10 ± 2.93 |
| Fully trainable | Yes | GCN | 90.74 ± 3.51 | 88.76 ± 4.43 | 2.95 ± 1.10 | 2.67 ± 0.97 |
| *(k = 16)* | Yes | MLP | 77.42 ± 10.63 | 70.40 ± 14.85 | 6.55 ± 2.89 | 6.21 ± 2.98 |
| | No | GCN | 92.91 ± 3.08 | 90.46 ± 4.08 | 2.32 ± 0.89 | 2.17 ± 0.84 |
| | No | MLP | 87.80 ± 5.07 | 84.27 ± 6.43 | 3.77 ± 1.42 | 3.55 ± 1.44 |

underperforming ResNet–18 (3D) with GCN+aug (92.76/90.48, MKE 2.31/2.19) by ≈1.1–1.6 Dice and ≈0.3 px MKE. In contrast, the video transformer *MViTv2–Small* achieves the best overall results with GCN+aug (93.05/90.84, MKE 2.26/2.14), edging both CNN baselines and ViT–Base. Transformers are more augmentation–sensitive: removing augmentation drops ViT–Base by $-2.46/-3.18$ Dice and adds ≈+0.68 px MKE, whereas MViT changes are modest ($-0.30/-0.49$ Dice). *Takeaway:* 3D temporal–aware transformers (e.g., MViT) with appropriate augmentation surpass 3D CNNs backbones.

**Decoder head impact.** Across backbones, GCN heads consistently outperform MLP heads (Table 2). With augmentation, GCN yields gains of roughly +0.8 to +1.8 Dice (ED) and +0.9 to +2.4 Dice (ES) and reduces MKE by ≈0.2–0.4 px; e.g., ResNet–18 (3D) improves from 91.66/89.05 to 92.76/90.48 (ED/ES) with MKE 2.62/2.48 → 2.31/2.19, and MViTv2–S from 91.29/88.40 to 93.05/90.84 with MKE 2.67/2.58 → 2.26/2.14. Effects remain in the no–aug setting for CNNs and MViT (e.g., ResNet–18 (3D): +1.28/ + 1.66 Dice), with one exception: ViT–Base without augmentation favors MLP (90.36/87.07 vs. 89.18/85.72). *Takeaway:* relational GCN decoding provides consistent, architecture–wide gains, especially for temporally aware encoders.

**Data augmentation effects.** Augmentation effects are architecture–dependent (Tables 2, 3). Transformers benefit the most: ViT–Base (GCN) gains +2.46/ + 3.18 Dice (ED/ES) and cuts MKE by ≈ 0.68 px (ED) and ≈ 0.68 px (ES) with augmentation, while MViTv2–S (GCN) improves by +0.30/ + 0.49 Dice and modest MKE drops (≈ 0.08–0.10 px). CNNs show smaller or mixed changes: ResNet–18 (3D, GCN) is nearly neutral (within ±0.1 Dice), and ResNet–18 (2D, GCN) slightly favors no–aug (+0.21/ + 0.23 Dice). *Takeaway:* Augmentation contributes to performance if partial or full training is applied.

### 3.2.2 Foundation model analysis

Table 3 shows that a *frozen* EchoPrime encoder underperforms and is unstable: with MLP heads it reaches $77.81 \pm 10.40/71.16 \pm 13.78$ (ED/ES, no augmentation) and with GCN $79.80 \pm 9.81/73.25 \pm 13.48$ (no augmentation), with high MKE ($\approx 6.3$–7.6 px) and large variance. Similar trends are also observed on the PanEcho model with MLP ($78.34 \pm 10.04/79.45 \pm 9.72$ for ED/ES without augmentation) and GCN $78.33 \pm 9.85/77.25 \pm 9.85$, no augmentation heads. These results indicate that fixed video–text features are misaligned with precise landmark localization. EchoPrime augmentation ablations mirror the trend observed for MViTv2–S: with a fully unfrozen encoder, augmentation markedly helps the GCN head (93.13 vs. 90.41 ED Dice; MKE 2.25 vs. 3.05), whereas under frozen/partial settings or with MLP heads, effects are neutral or negative. Allowing the encoder to adapt closes the gap rapidly: unfreezing just two blocks ($k=2$) with GCN+aug yields $91.71 \pm 3.49/89.05 \pm 4.61$ and MKE 2.63/2.49, while fully trainable encoder ($k=16$) with GCN+aug reaches $93.13 \pm 3.11/90.95 \pm 3.71$ and MKE 2.25/2.11, on par with or exceeding the best non–foundation baselines (Table 2). *Takeaway:* while EchoPrime and PanEcho are less effective as an off-the-shelf fixed feature extractor, they become more useful for landmark detection as backbones that are selectively

**Table 4. Experiment on limited data (10%):** Performance comparison of *frozen* and *unfrozen* models using EchoPrime, MViTv2–S, and PanEcho backbones. We report mean ± std Dice at ED/ES [%]. All models use 16–frame input sequences with augmentations; heads are GCN unless otherwise noted.

| Model | Dice ED ↑ | Dice ES ↑ |
|---|---|---|
| *Frozen (k = 0)* | | |
| EchoPrime (GCN) | $74.91 \pm 10.92$ | $68.06 \pm 14.15$ |
| MViTv2–S (GCN) | $75.89 \pm 10.64$ | $68.75 \pm 14.15$ |
| PanEcho (GCN) | $74.81 \pm 11.09$ | $68.25 \pm 14.62$ |
| *2 blocks unfrozen (k = 2)* | | |
| EchoPrime (GCN) | $87.81 \pm 5.56$ | $83.96 \pm 7.29$ |
| MViTv2–S (GCN) | $87.20 \pm 5.96$ | $83.13 \pm 7.77$ |
| *Fully trainable (k = 16)* | | |
| EchoPrime (GCN) | $90.34 \pm 4.37$ | $87.40 \pm 5.35$ |
| MViTv2–S (GCN) | $89.54 \pm 4.67$ | $86.38 \pm 5.87$ |
| PanEcho (GCN) | $89.61 \pm 4.30$ | $85.87 \pm 5.53$ |

fine–tuned; small unfreezing already captures most gains, and full unfreezing maximizes accuracy.

**Critical impact of partial fine–tuning.** Unfreezing only two encoder blocks ($k=2$) captures most of the benefit. With MLP (no augmentation), Dice rises from 77.81/71.16 (ED/ES, $k=0$) to 91.40/88.41, while MKE drops from 6.93/6.61 to 2.70/2.61. With GCN+aug, the jump is similarly large: 78.41/71.96 → 91.71/89.05 with MKE 6.81/6.50 → 2.63/2.49. The remaining gap to fully trainable (full unfreezing) is modest: relative to $k=16$ (GCN+aug), $k=2$ is within 1.42 (ED) and 1.90 (ES) Dice. *Takeaway:* selective fine–tuning with small $k$ yields Dice *close to full training* (within ~1–2 points).

**Full training of FMs: Does the pretraining help?** With full encoder training ($k=16$), EchoPrime with GCN+aug reaches 93.13/90.95 Dice (ED/ES) and MKE 2.25/2.11 (Table 3). This performance exceeds all non–foundation backbones ( Table 2) while demonstrating superior consistency (low standard deviations). It is also worth noting here that EchoPrime with GCN + Aug achieves the *best performance over all existing models on the EchoNet benchmark based on landmark regression.* While this EchoPrime configuration was able to get record breaking results, we also observe that the MViTv2-S, pretrained on the Kinetics–400 action dataset, with GCN+aug, achieves very close performance, namely 93.05/90.84 Dice (ED/ES).

A plausible explanation for the gains with full or partial parameter training might be attributed to the specific phase alignment in our data: ED is always frame 0 and ES frame 15, whereas EchoPrime was pre–trained across diverse cardiac cycles and phases. This mismatch can affect early tokenization and the encoder's temporal inductive biases, making frozen features suboptimal for precise landmark localization. Training more parameters enables the encoder to adapt its representations to better fit

the specific temporal structure required by the task, improving both accuracy and stability.

### 3.2.3 Limited data performance

With only 10% of the training data (Table 4), unfreezing drives most of the gains for both EchoPrime and MViTv2–S. From frozen ($k$=0) to partially unfrozen ($k$=2), Dice improves by a +12 to +16 points while the variance halves (e.g., EchoPrime ED: 74.91±10.92 → 87.81±5.56; ES: 68.06±14.15 → 83.96 ± 7.29). Full parameter training ($k$=16) adds a smaller but consistent boost (EchoPrime ED/ES: 90.34/87.40; MViTv2–S: 89.54/86.38), retaining ∼95–97% of each model's full–data accuracy. Notably, $k$=2 already captures ≈82–84% of the improvement from frozen to fully unfrozen for both backbones. These trends align with the data–efficiency curves (Fig. 2), where the largest gains appear between 0.5–25% of data and top models reach around 0.90 Dice (ED) by the 10% split, with 10% runs completing in ∼1.5 hours. *Takeaway:* in low–data regimes, modest unfreezing delivers most of the accuracy and stability benefits, while full parameter training provides the fine (smaller) improvements.

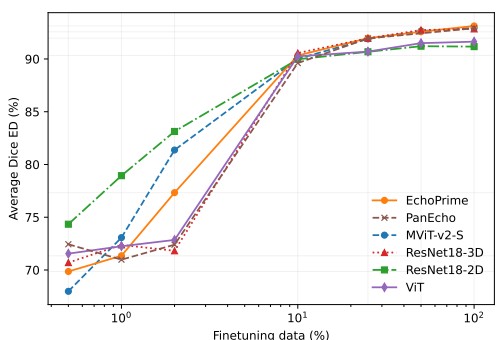

**Figure 2. Data efficiency across five backbones.** We compare ResNet-18 (2D) (GCN), ResNet-18 (3D) (GCN), EchoPrime (GCN), MViTv2-Small (GCN), and ViT-Base (GCN). $x$-axis: training-set fraction (0.5%, 1%, 2%, 10%, 25%, 50%, 100%) on a logarithmic scale; $y$-axis: Average Dice ED (112) [%]. All the models were trained end-to-end with all blocks unfrozen.

### 3.2.4 Impact of frozen layers

Figure A.1 (see Appendix A) summarizes how accuracy and memory scale with the number of unfrozen blocks $k$ at 100%, 25%, 10%, and 2% data. For both EchoPrime (solid) and MViTv2–S (dashed), Dice rises steeply from $k$=0 to $k$≈4, after which gains slow down, especially at 2–10% data, while at 100% a small additional improvement persists up to $k$≥12. As expected, we also show how GPU memory increases gradually with $k$, so small $k$ captures most of the accuracy at substantially lower memory. These trends align with Tables 3 and 4: partial fine–tuning

yields most of the benefit, whereas full training with augmentation achieves the best result.

## 4 Discussion & Conclusion

We assessed the suitability of public echocardiography FMs for spatio–temporal landmark detection and found that although they show impressive performance on other downstream tasks, *encoder adaptation is essential* for this particular complex task. Frozen encoders perform poorly and variably, while selectively unfreezing only two blocks ($k$=2) already recovers most of the accuracy and stability gains; full parameter training provides the final, smaller improvements (Table 3, Table 4). GCN heads consistently outperform MLP across backbones, indicating that modeling anatomical relations between landmarks is beneficial (Table 2). Among non–FM baselines, MViTv2–Small and ResNet–18 (3D) are strong, and fully trainable EchoPrime with GCN+augmentation is competitive under our ED/ES protocol.

Practically, the accuracy–compute trade–off is favorable at small unfreezing budgets: $k$≈2 captures most gains at lower memory, while $k$≥12 is best reserved for maximal accuracy (Fig. A.1). Data–efficiency analyses show that improvements concentrate at small fractions (0.5–25%), with 0.5–1% runs completing in ∼10–15 minutes (Fig. 2). These trends are consistent with our fixed–phase setup (ED at frame 0, ES at frame 15): fine–tuning likely realigns pretraining features, learned across diverse cardiac phases, to the downstream task's temporal structure.

Overall, the echocardiography FMs explored here have demonstrated strong performance across diverse downstream tasks; in our landmark–detection setting, however, adopting FMs for precise localization tasks requires additional training and parameters. Selective fine–tuning, paired with a graph decoder, unlocks most of the benefit, making FM such as EchoPrime [4] an effective and resource–efficient backbone for this task. For resource–constrained use, partial unfreezing (e.g., $k$≈2) with augmentation and a GCN head has shown to be most useful. Future work should explore parameter–efficient adaptation (e.g., low–rank adapters), broader datasets and protocols in which the network can infer cardiac phase directly, wider range of FMs (e.g. EchoFM), robustness across vendors and acquisition settings, and unified comparisons on identical evaluation criteria.

## Acknowledgments

This work was funded by the Research Council of Norway through Visual Intelligence, Centre for Research-based Innovation (309439).

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

# A Appendix

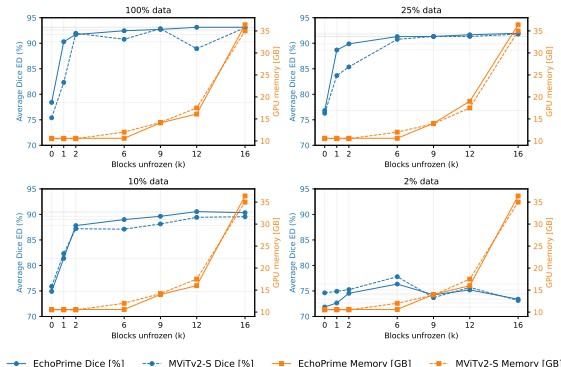

**Figure A.1. Effect of backbone unfreezing depth ($k$) on segmentation performance and GPU memory usage across dataset sizes.** Each panel shows results for a different fraction of the training set (100%, 25%, 10%, 2%). Left $y$-axis (blue): Average Dice ED (112) [%]. Right $y$-axis (orange): GPU memory usage [GB]. Solid lines: EchoPrime backbone; dashed lines: MViTv2-S backbone. Compute time for full training varies substantially: ~13 hours (100%), ~3.5 hours (25%), ~1.5 hours (10%), and ~0.25 hours (2%).

