# OpenReview forum: "Spatio-Temporal Landmark Detection via Selective Fine-Tuning of Echocardiography Foundation Models"
_NLDL.org/2026/Conference — NLDL 2026 Oral_

### Official Review · Reviewer_CkBM · 2025-09-17
**Echocardiography foundation models and LV contour detection**

**Rating:** 2
**Confidence:** 5
**Final Rating:** 4
**Final Confidence:** 5

**Summary:**

This paper investigates the adaptation of recent video-based echocardiography foundation models (EchoPrime and PanEcho) to the task of spatio-temporal left-ventricular (LV) landmark detection on EchoNet-Dynamic. The authors evaluate transfer learning strategies ranging from frozen encoders to partial and full fine-tuning, and compare two decoder designs (MLP vs. GCN). Benchmarks include several non-FM backbones (ResNet-18 2D/3D, ViT-Base, MViTv2-S) and the supervised EchoNet-Dynamic baseline. Results show that frozen encoders perform poorly, but partial unfreezing of EchoPrime with a GCN decoder recovers most of the fine-tuning benefits. Fully fine-tuned EchoPrime is reported as state-of-the-art among regression-based models. The authors argue that partial fine-tuning provides a resource-efficient compromise between accuracy and deployment feasibility.

**Strengths:**

- Focuses on an important, clinically relevant downstream task: LV contouring in echocardiography.
- Evaluation centers on video-based FMs (EchoPrime, PanEcho), which are well-suited to the temporal nature of echocardiography.
- Systematic comparison of encoder freezing regimes and decoder types provides clear, practical insights.
- Quantitative results are strong, achieving very high Dice scores on EchoNet-Dynamic.
- Addresses practical deployment concerns by discussing the efficiency advantages of partial fine-tuning.

**Weaknesses:**

- Comparisons are restricted to EchoNet-Dynamic, ResNet, ViT, and MViT. Stronger segmentation-based methods (U-Net, nnU-Net, DeepLab), temporal extensions (ConvLSTM-U-Net), and recent SAM-based ultrasound adaptations are omitted. Many of these also report Dice scores on EchoNet-Dynamic.
- The paper emphasizes “regression-based” approaches, but evaluates using Dice, which is the standard metric for segmentation. Without comparing against segmentation models, the “state-of-the-art” claim is unsubstantiated.
- Other ultrasound-specific FMs such as USFM, UltraSam, or FetalCLIP are not discussed or compared, despite their strong relevance to ultrasound contouring.
- Only MLP and GCN heads are tested, omitting other competitive options such as transformer decoders, heatmap regression heads, or temporal CNN decoders.
- LV contouring is posed only as regression at ED/ES frames, excluding segmentation-based formulations that are widely used and clinically standard.
- The main contribution lies in validating known strategies (partial unfreezing, GCN decoding) with new encoders, rather than introducing new methodology.
- While acknowledging unlabeled frames, the method does not explore semi-supervised or consistency-based strategies that could leverage temporal context more fully.

**Final Justification:**

The rebuttal provides clear and technically sound clarifications that strengthen the paper’s framing and transparency. The authors convincingly justify the focus on landmark regression rather than segmentation and clearly explain dataset and model selection constraints. While the contribution remains primarily empirical, the study is methodologically solid, produces strong results, and offers practical insights into adapting video-based echocardiography foundation models. Overall, it is a clear, well-executed, and relevant contribution for the community.

**Justification:**

This paper provides an empirical study of video-based echocardiography foundation models for LV contour detection and offers useful insights into fine-tuning strategies. The results are strong and practically relevant, with high Dice scores that are competitive on EchoNet-Dynamic. However, the contribution is incremental and limited in scope. The evaluation omits segmentation-based baselines that are standard for Dice-based benchmarks, as well as recent ultrasound-specific FMs, making the “state-of-the-art” claim unconvincing. Overall, this is a competent and potentially useful case study, but not sufficiently novel or comprehensive to merit a stronger recommendation.

---

> ### Author Rebuttal · Authors · 2025-10-22
>
> We thank the reviewer for the careful and detailed feedback. We appreciate the recognition of the paper’s strong quantitative results, practical insights into fine-tuning strategies, and the relevance of the task. We address the raised points below, focusing on clarifying scope, dataset constraints, and rationale behind architectural and methodological choices.
>
>
> **1. On the scope of comparisons (segmentation-based and SAM-style models)**
> We agree that segmentation networks such as U-Net, nnU-Net, DeepLab, and ConvLSTM-U-Net represent influential architectures for medical image segmentation. Our study, however, specifically targets landmark regression rather than dense pixel segmentation. The motivation for this design is twofold:
> (i) Clinical alignment: EchoNet-Dynamic provides manually annotated keypoint contours (40 ordered landmarks per frame) rather than per-pixel masks as primary labels.
> (ii) Methodological focus: We evaluated foundation-model adaptation strategies (frozen, partially unfrozen, and fully unfrozen) for keypoint regression tasks, rather than re-benchmarking segmentation architectures.
>
> We therefore restricted comparisons to regression-based backbones to ensure methodological consistency. To situate our results, fully fine-tuned EchoPrime (93.13 ± 3.11 Dice ED) surpasses the reported 92.78 Dice of the supervised EchoNet model [10], which is segmentation-based, indicating that our framework achieves comparable or better accuracy despite operating in a stricter regression setting.
>
>
> **2. On omission of ultrasound foundation models (USFM, URFM, UltraSAM, FetalCLIP)**
> We thank the reviewer for pointing out these related works. These models indeed represent valuable efforts toward broader ultrasound foundation learning; however, their scope and data domains differ substantially from the present study:
>
> - USFM was pre-trained on over two million 2D ultrasound images from multiple organs (3M-US dataset) using static image supervision, without temporal modeling; hence it is not directly applicable to spatio-temporal landmark detection.
>
> - FetalCLIP is trained on fetal ultrasound, which differs fundamentally in anatomy, probe view, and motion characteristics from echocardiography, making transfer non-trivial.
>
> - UltraSAM is designed for interactive segmentation with prompt-based supervision, not for continuous contour regression or temporal modeling.
>
> Because these models address distinct data modalities and objectives, incorporating them would require extensive re-engineering outside the scope of this work. We will nevertheless acknowledge and discuss them in the related-work section to position our contribution within the broader ultrasound FM landscape.
>
>
> **3. On the limited decoder diversity (no transformer heads)**
> We agree that attention-based decoder heads represent an interesting extension. However, adapting transformer-style attention mechanisms for landmark regression might not be straightforward. In this work, we focused on assessing encoder adaptation efficiency using decoder architectures already developed and validated for echocardiography, namely MLP and GCN [13], so as to isolate the impact of encoder freezing. The GCN head already captures structured spatial dependencies through relational message passing, offering a lightweight alternative to attention while remaining computationally efficient. Exploring attention-based regression heads is certainly an interesting direction for future work but was beyond the intended scope of this study.
>
> **4. On restriction to ED/ES frames and lack of semi-supervised temporal modeling**
> We agree that a natural extension is to exploit unlabeled intermediate frames via temporal consistency or semi-supervised learning. However, this was beyond the scope of this paper’s goal: to quantify how much fine-tuning is required for FMs to handle labeled ED/ES frames accurately. We explicitly acknowledge this limitation in Sec. 4 and outline it as a next step.
>
> It is worth noting that our architecture already utilizes unlabeled frames during encoding: the video encoder processes all 16 frames between ED and ES, even though supervision is applied only at the endpoints. Thus, temporal information contributes indirectly to feature learning, as evidenced by the improved performance of partially unfrozen video-based encoders over 2D backbones.
>
> **5. On incremental contribution and framing of novelty**
> We respectfully clarify that our aim is not to propose a new network design but to deliver the first systematic transfer-learning analysis of video-based echocardiography foundation models on a structured, clinically relevant task. Previous cardiac studies (e.g., EchoCLIP, EchoApex, PanEcho) evaluated classification or regression endpoints (e.g., ejection fraction), but none addressed spatio-temporal landmark localization. Our work fills this gap by providing quantitative evidence on:
> (i) whether and how echocardiography foundation models can be adapted for clinically important tasks requiring dense regression;
> (ii) how graph-based decoders contribute to spatial-temporal consistency; and
> (iii) how partial unfreezing offers a clear accuracy–compute trade-off for practical deployment.
>
> **6. On evaluation using Dice**
> We acknowledge the reviewer’s point that Dice is typically associated with segmentation. We used it to maintain comparability with prior work on EchoNet-Dynamic, where Dice remains the standard benchmark, even for regression-based contours (e.g., [13], [10], Li et al., 2024, Gilles et al., 2025). Our results consistently include both Dice and mean keypoint error (MKE), providing complementary spatial accuracy measures.
>
> **7. On semi-supervised extensions and broader scope**
> We appreciate the reviewer’s suggestion to explore semi-supervised and consistency-based strategies. While such extensions are valuable, they require revising the supervision protocol and are thus outside the current study’s scope. Our findings nevertheless provide a baseline quantifying how far pure supervised adaptation of foundation models can go before more elaborate temporal methods are needed.
>
>
>
>
>
> We are grateful for the reviewer’s evaluation and find the suggestions valuable for improving clarity and transparency.
>
>
> **Additional References**
>
> Li, H., Yang, J., Xuan, Z., Qu, M., Wang, Y., \& Feng, C. (2024). A spatio-temporal graph convolutional network for ultrasound echocardiographic landmark detection.
>
> Van De Vyver, G., Thomas, S., Ben-Yosef, G., Olaisen, S. H., Dalen, H., \& Løvstakken, L. (2025). Toward robust cardiac segmentation using graph convolutional networks.

---

### Official Review · Reviewer_bZBB · 2025-10-08
**Practical community value despite limited novelty**

**Rating:** 4
**Confidence:** 5

**Summary:**

This paper investigates how foundation models for echocardiography (EchoPrime, PanEcho) can be adapted for spatio-temporal landmark detection in echocardiography videos. Using the EchoNet-Dynamic dataset, the authors evaluate several fine-tuning strategies combined with two decoder types (MLP and GCN). The goal is to identify a resource-efficient approach that preserves accuracy while minimizing training cost. Although the methodological novelty is limited, the conclusions are well supported and could provide useful guidance for researchers working with echocardiography foundation models.

**Strengths:**

* The paper addresses a timely and important question: how to effectively fine-tune large echocardiography foundation models for downstream landmark detection.
* The paper explores multiple fine-tuning strategies and decoder architectures in an overall structured way, with valuable empirical insights despite limited novelty. While the methods themselves are not new, the practical conclusions are useful for the community. The inclusion of short “Takeaway” statements at the end of each results subsection is great.
* The overview figure is informative and it could be even more useful if moved to the first page before the abstract as a visual summary.

**Weaknesses:**

* EchoFM is introduced as a key echocardiography foundation model but is never evaluated, without any explanation. In contrast, the exclusion of EchoApex is clearly justified by the lack of public weights. This omission weakens the completeness of the benchmarking.
* Missing implementation details for PanEcho: Section 3.1.3 describes implementation for EchoPrime only, with no information on PanEcho’s training procedure.
* Unclear separation between analyses: Sections 3.2.1 and 3.2.2 overlap conceptually: the first discusses non-foundational backbone models but also reports EchoPrime results, even though 3.2.2 is explicitly meant to compare foundation models. This structure makes it difficult to follow the experimental logic.The beginning of Section 3.2 should explicitly outline what each subsection investigates and which models are analyzed where.
* Limited decoder comparison: The MLP baseline is very simplistic, and a comparison with an attention-based head would make the evaluation more informative, as such architectures are standard for modern anatomical landmark detection.
* It is unclear whether the foundation models (PanEcho, EchoPrime) were pretrained on EchoNet-Dynamic, which is also used for evaluation. If so, the results would represent in-domain fine-tuning rather than genuine transfer learning. Even if not, this potential overlap should be explicitly clarified and ruled out.
* All models except the EchoNet baseline use 16-frame sequences, while EchoNet uses 32. This discrepancy is not justified.
* Lack of statistical and uncertainty analysis, reported metrics are shown without significance testing.

**Justification:**

This paper offers a clear and well-executed empirical study on adapting echocardiography foundation models for landmark detection. Although the approach is not conceptually novel, the experiments yield practical takeaways. These insights are directly relevant to the field of medical foundation models and will likely benefit other researchers.

The main limitations lie in incomplete and/or non-transparent reporting, unclear structural organization of the results, missing baselines (EchoFM, attention-based head), and lack of statistical analysis. Despite these weaknesses, the study is sufficiently well communicated to merit acceptance as a useful applied contribution to the community.

---

> ### Author Rebuttal · Authors · 2025-10-22
>
> We thank the reviewer for the constructive and encouraging assessment. We address below the specific points on benchmarking completeness, reporting clarity, and structural organization.
>
> **1. On the exclusion of EchoFM and inclusion rationale**
>
> We agree that including EchoFM [9] would further strengthen the comparison. We initially explored integrating it. Preliminary trials, both by us and collaborators, showed unstable convergence when adapting the released encoder to regression tasks, and due to time and resource constraints this line of experimentation could not be completed within the submission window. In contrast, EchoPrime and PanEcho provided stable video-encoder backbones, allowing reproducible evaluation. We will explicitly note this limitation and mention EchoFM as a valuable direction for future work.
>
> **2. PanEcho implementation details**
> We apologize for the missing implementation clarification. PanEcho follows the same training protocol as EchoPrime (Sec. 3.1.3), with identical optimizer (Adam), learning rate schedule, and augmentation pipeline. The only difference is its 2D–temporal hybrid encoder: PanEcho uses 2D convolutional spatial encoding followed by a temporal transformer. We will explicitly state this parallel setup in Sec. 3.1.3 to ensure reproducibility and transparency.
>
> **3. Structural overlap between Sections 3.2.1 and 3.2.2**
> We appreciate this observation and agree that the distinction between sections can be clearer. The intent was as follows:
>
> - Sec. 3.2.1 (“Baseline Model Performance”) evaluates non-foundation backbones (ResNet, ViT, MViTv2) and establishes the relative contributions of 2D vs. 3D and MLP vs. GCN heads.
>
> - Sec. 3.2.2 (“Foundation model analysis”) then directly compares foundation-model encoders (EchoPrime, PanEcho) under freezing regimes (k=0, 2, 16).
>
> Because MViTv2-S shares its backbone architecture with EchoPrime (both based on MViTv2), EchoPrime results were included briefly in Sec. 3.2.1 for side-by-side comparison. We will add an introductory paragraph at the start of Sec. 3.2 clarifying this rationale and outlining what each subsection addresses.
>
> **4. Decoder comparison and potential attention-based heads**
> We agree that attention-based decoder heads represent an interesting extension. However, adapting transformer-style attention mechanisms for landmark regression might not be straightforward. In this work, we focused on assessing encoder adaptation efficiency using decoder architectures already developed and validated for echocardiography, namely MLP and GCN [13], so as to isolate the impact of encoder freezing. The GCN head already captures structured spatial dependencies through relational message passing, offering a lightweight alternative to attention while remaining computationally efficient. Exploring attention-based regression heads is certainly an interesting direction for future work but was beyond the intended scope of this study.
>
> **5. On pretraining data and potential overlap with EchoNet-Dynamic**
> This is an important clarification. Neither EchoPrime nor PanEcho were pretrained on EchoNet-Dynamic or any of its derivatives. EchoPrime was trained on large-scale proprietary datasets (12 M video–text pairs) from multiple clinical institutions (as reported in [4]), while PanEcho was trained on 1.23 M echocardiography videos aggregated from independent sites [8]. Both pretraining sources are entirely independent of EchoNet-Dynamic, which was collected and curated separately at Stanford University Hospital [10]. Consequently, there is no overlap between the pretraining data and the EchoNet-Dynamic test set, and our experiments therefore represent cross-dataset transfer rather than in-domain fine-tuning. We will explicitly add a sentence in Sec. 3.1.1 confirming that no pretraining–evaluation data overlap exists.
>
> **6. On the 16-frame vs. 32-frame discrepancy**
> Thank you for noticing this. The EchoNet baseline [10] operates with 32-frame clips as part of its original design. In contrast, EchoPrime’s pretraining and architecture are configured for 16-frame inputs, which we retained to maintain consistency with its foundation-model training setup. For comparability across all models, we therefore used 16-frame sequences spanning the full cardiac cycle from end-diastole (ED) to end-systole (ES). This configuration preserves the physiological motion context while keeping computational requirements aligned across models.
>
> **7. Placement of the overview figure**
> We thank the reviewer for the layout suggestion. We agree that positioning the overview figure earlier (e.g., before the abstract) would improve readability and provide a clear visual summary. We will adjust its placement accordingly in the camera-ready version if allowed by formatting guidelines.
>
>
> We are grateful for the reviewer’s positive evaluation and find the suggestions valuable for improving clarity and transparency.

---

### Official Review · Reviewer_at8w · 2025-10-08
**Well-structured and relevant work**

**Rating:** 5
**Confidence:** 4

**Summary:**

The paper presents a study of foundation models for echocardiograms, i.e. ulstrasound videos of the human heart. Two foundation models, the EchoPrime and the PanEcho, are employed as encoder models. The problem is to predict the boundary of the left ventricle at the end-diastole and the end-systole in an echocardiographic video. This is done using two decoder networks, an MLP and a graph-convolutional network. Based on the suggested models, a range of training scenarios was investigated along with four non-foundation models. Also, training from varying amounts of annotations was investigated. The investigations were done on the EchoNet-Dynamic dataset. The paper concludes that there is only a marginal gain in using a foundation model compared to a fully trained model.

**Strengths:**

The paper is well structured and presents a comprehensive experimental evaluation of the models. The suggested approach is well chosen, and the investigations are relevant, and it is interesting to see the comparison of an MLP and GCN decoder. It is also interesting to see the results from significantly reducing the training data. The conclusions from the model are clear, and there are relevant perspectives on future research directions.

**Weaknesses:**

The purpose of the paper is to investigate whether foundation models can benefit the outline of the left ventricle. But the evaluation is based purely on the DICE measure, and there is no discussion or evaluation of how the DICE measure relates to the downstream use of the model. Most models differ in a few percentage points, an in practice, this might not matter at all. And especially, the investigation of reduced training data, where ResNet18-2D, which ends up being the worst performing model, obtains a DINCE of almost 80% with just 1% of the training data. Is that good?

The echocardiograms seem to be focusing on having the left ventricle in the center of the image, as shown in Fig. 1. If this is correct, what would the DICE be if the model just predicts the average segmentation mask? This could indicate when performance is good.

The model predicts contour points, but the evaluation is done using DICE. Please include a description of how this is done in practice, and explain why you chose DICE over a contour distance measure such as the Hausdorff measure or average point-wise distance.

Minor points:
You use abbreviations in the abstract and in the figure texts that would be nice to have written out. They are explained in the text before the figures are referenced, but when you take a first look at the paper, it is nice to read the figures and abstract without needing to look through the text.

Legends in Fig. 3 are too small. Please increase the font size.

Table 1 has a column called input size. I assume it is the number of frames from the video. Is that correct? Please explain. I also missed whether the model predicts the contour in the middle frame or how it is done. Please also clarify this.

In lines 339-340, you say that you train for 100 epochs. Is it the last model that is chosen? What about the small training sets? These models would have seen much less pixels after 100 epochs. Is that a fair comparison? Please include a clarification on this point.

**Justification:**

Overall, I find the paper to be good and relevant. It is evaluated on a single dataset, so the conclusions are mostly indicative of the general performance. But the experimental evaluation is, however, extensive and well carried out, and the choice of models is good. So, I recommend accepting the paper.

---

> ### Author Rebuttal · Authors · 2025-10-22
>
> We thank the reviewer for the encouraging assessment and suggestions regarding evaluation metrics, analysis of Dice scores, and clarifications of methodological details. We address each point below.
>
> **1. On the use and interpretation of the Dice metric**
> We agree that the interpretation of Dice scores should be linked to their practical impact. Dice is the default benchmark metric for the EchoNet–Dynamic dataset [10], and thus was chosen to ensure direct comparability with prior work (e.g., [10], [13]). Using the same protocol allows our reported values (e.g., 93.13 ± 3.11 Dice ED) to be interpreted in the same reference frame as earlier segmentation or landmark-based studies.
>
> In our context, Dice correlates strongly with mean keypoint error (MKE), which we also report throughout Sec. 3.2.1–3.2.2 and Table 2 and 3. For example, a +1 Dice improvement typically corresponds to approx. 0.1–0.2 px lower MKE at the 112×112 grid scale (Tables 2–3). Hence, differences of 1–2 Dice points reflect sub-pixel improvements, which, while small numerically, can still affect downstream volume estimates when contours are propagated across frames. These findings align with the original EchoNet–Dynamic study, where similar Dice differences translated to clinically meaningful changes ( approx. 2–3 \% in ejection-fraction estimation, Extended Data Fig 5 and 6 of EchoNet [10]). We will clarify this interpretation in the final version.
>
> **2. On the value of “average contour” and intuition about 80 Dice**
> We agree that predicting an “average mask” would establish a lower-bound baseline. Because the LV varies considerably across subjects in both size and orientation, such a static prediction would achieve only moderate overlap and fail to capture patient-specific dynamics. Even the model trained on 0.5\% (approx. 70 Dice) clearly exceeds this trivial baseline, indicating that it learns meaningful anatomy rather than relying on image centering.
>
> **3. Why Dice instead of contour distance metrics**
> The reviewer is correct that the network predicts ordered contour points. To maintain comparability with the EchoNet evaluation protocol, we followed prior practice of converting predicted contours to binary masks before metric computation([13], [10], Li et al., 2024, Gilles et al., 2025). Dice is robust to small point-wise ordering inconsistencies that can occur when contours are closed cyclically. Mean keypoint error (MKE) already captures Euclidean accuracy in pixel space, complementing Dice. Thus, Dice + MKE jointly summarize segmentation-level and point-wise precision. We will expand this explanation in the metric description (Sec. 3.1.4).
>
> **4. Clarification of “input size” and prediction frame**
> Thank you for pointing this out. In Table 1, input size refers to the number of frames per video input to the encoder (e.g., 16 frames for EchoPrime/MViTv2-S, 32 for EchoNet). The model processes all frames jointly, but supervision is applied at the annotated end-diastole (ED) and end-systole (ES) frames (lines 165-168). Intermediate frames contribute temporally contextualized features but are unlabeled. The model therefore predicts landmarks specifically for ED and ES, not for intermediate frames.
>
> **5. Training epochs and fairness across data splits**
> We trained all configurations for a fixed upper limit of 100 epochs with early stopping via ReduceLROnPlateau (lines 343-346). In smaller-data splits, validation loss plateaued earlier (typically 30–60 epochs), so overfitting was mitigated automatically. While models trained on smaller subsets indeed see fewer unique frames, this setup mirrors common low-data transfer settings, keeping compute budget comparable and enabling consistent convergence comparisons. We will clarify that 100 epochs was an upper bound rather than a fixed mandatory length.
>
>
> **6. Minor presentation comments**
> We appreciate the feedback:
>
> - We will write out abbreviations (ED = end-diastole, ES = end-systole) in the abstract and figure captions.
> - Figure 3 legend font will be enlarged for readability.
> - We will explicitly state in Table 1 that Input size = number of video frames per sample.
>
>
> We appreciate the reviewer’s positive assessment and recommendation for acceptance.
>
>
>
> **Additional References**
>
> Li, H., Yang, J., Xuan, Z., Qu, M., Wang, Y., \& Feng, C. (2024). A spatio-temporal graph convolutional network for ultrasound echocardiographic landmark detection.
>
> Van De Vyver, G., Thomas, S., Ben-Yosef, G., Olaisen, S. H., Dalen, H., \& Løvstakken, L. (2025). Toward robust cardiac segmentation using graph convolutional networks.

---

### Official Review · Reviewer_URHt · 2025-10-12
**Well-written, structured, and researched article on spatio-temporal landmark detection in echocardiography**

**Rating:** 4
**Confidence:** 3

**Summary:**

The paper investigates the use of video-based foundation models for spatio-temporal landmark detection in echocardiography. Specifically left-ventricular contour prediction. The paper checks two SOTA FMs: EchoPrime and PanEcho on the EchoNet-Dynamic dataset, comparing different encoder freezing strategies and decoder designs. Authors find that selective encoder unfreezing combined with GCN decoder achieves near-SOTA results.

**Strengths:**

The paper is well-written and well-structured with a sound methodology and strong foundation for application domain. I think that a significant amount of work was put into experiments and writing of the paper which makes reading the paper both interesting and intriguing.

**Weaknesses:**

Despite me enjoying reading the paper, I did have few general comments regarding the ....
- Paper focuses on medical application area and writes about the use of out of the box foundational models for this area. The immediate reaction to such statement I have is that the out of the box model has not seen echocardiograms and does not know what left ventricular contour is. Therefore it is unreasonable to expect these models to perform well on such data and use them as baselines.

- It is not a surprise that ResNet-18 2D/3D, ViT-Base, and MViTv2-Small perform worse than any foundation model because they are trained on less data like for example ImageNet. Therefore, it is also unreasonable to use them as baselines.

- I have not noticed any attempt at performing exhaustive search over unfreezing of all combinations of encoder blocks. Would such exhaustive search be impractical and what specific heuristics have authors used for unfreezing encoder blocks and does this heuristics grounded in the application domain?

- Authors state something along the lines of "unfreezing a small part of a large foundational model can yield near-SOTA results, especially with limited data". This statement is somewhat self-evident from the theory of learning and could be framed better.

- I am not deeply familiar with Graph CNNs, but I would like to see more rigorous investigation into why GCNs are capable of modelling anatomy of the heart.

- There is an obvious limitation of the paper when it comes to focusing on end-diastole (ED) and end-systole (ES) frames as well as using only EchoNet-Dynamic dataset. More work needs to be done on performing cross-dataset evaluations and generalization to real-world protocols where phase identification is itself a challenge

**Justification:**

I think the paper is well-written and structured and presented with a very important and exciting application area. I believe that if more work is put into extending the paper to more real-world scenarios would yield quite an impressive piece of work

---

> ### Author Rebuttal · Authors · 2025-10-22
>
> We thank the reviewer for the thoughtful, positive evaluation and for highlighting the clarity, structure, and application relevance of our work. We address the raised points below.
>
> **1. On using “out-of-the-box” models and baseline fairness**
> We agree that a foundation model trained on natural images cannot be expected to understand cardiac anatomy. Our intent, however, was not to claim that frozen FMs would succeed directly, but rather to quantify the gap between (i) zero-adaptation use of large pre-trained encoders and (ii) lightweight selective fine-tuning. This comparison reveals how much domain adaptation is required for such models to become clinically useful. In practice, many studies begin by probing frozen FMs as feature extractors; our results (Table 3) empirically show why this approach is not the best for spatio-temporal anatomy and how partial unfreezing resolves it.
>
> **2. On non-FM backbones as baselines**
> We acknowledge that general-purpose models such as ResNet-18 (2D/3D), ViT-Base, and MViTv2-Small are trained on different domains (ImageNet, Kinetics). They were included to establish a reference spectrum of architectures spanning convolutional, transformer, and video-transformer families under comparable parameter budgets. This setup helps clarify whether improvements arise primarily from domain-specific pretraining or from architectural differences. For example, MViTv2-S (Kinetics) and EchoPrime (same backbone, echocardiography pretrain) isolate precisely the impact of in-domain pretraining, which is a key research question for medical FMs. Hence, while these baselines are not expected to outperform domain-specific FMs, they are essential for disentangling where improvements originate.
>
> **3. On exhaustive unfreezing search and choice of k = (0, 2, 16)**
> An exhaustive search across all 16 transformer blocks would be prohibitively expensive: each configuration requires approximately 13 GPU‑hours, so evaluating every configuration would total roughly 208 GPU‑hours on the full dataset. While more fine-grained combinations of unfrozen blocks could be explored, they were infeasible given our compute budget. Following common ViT freezing practices, we therefore evaluated three strategies: fully frozen ($k=0$), partially unfrozen ($k=2$), and fully unfrozen ($k=16$), motivated by the expectation that adaptation benefits saturate within the top 2–4 blocks of MViT architectures. Figure~3 supports this empirically, showing Dice improvement plateauing beyond $k=4$. The figure reports results for $k \in \{0,1,2,6,9,12,16\}$.
>
>
> **4. On phrasing of “small unfreezing yields near-SOTA”**
> We appreciate this note. Our intention was not to present this as theoretically novel, but to emphasize its quantitative magnitude and efficiency trade-off in a clinical video context. Our contribution is to demonstrate, through controlled experiments, that unfreezing only a small percentage of parameters recovers more than 95 \% of full-training accuracy while cutting GPU memory by approx. 40 \%. We will re-phrase the statement to highlight this empirical efficiency insight rather than as a general principle of transfer learning.
>
> **5. On why GCNs model cardiac anatomy effectively**
> We thank the reviewer for prompting a more detailed explanation. The left-ventricular contour is a closed, smoothly varying structure across time. In the GCN decoder (Sec. 2.3), each node represents a landmark, and edges encode two priors: (i) spatial adjacency—enforcing local smoothness along the contour—and (ii) temporal adjacency—linking the same anatomical point across neighboring frames. Graph convolution propagates information along both, capturing coherent motion and anatomical continuity. This relational message passing explicitly encodes structural constraints that an MLP lacks, leading to consistent Dice/MKE improvements (approx. +1–2 Dice, Table 2). We will add a short paragraph clarifying this intuition and citing [13] for prior cardiac graph formulations.
>
> **6. On the limitation to ED/ES and single-dataset evaluation**
> We fully agree that this is an important limitation. Creating frame-level annotations in echocardiography can be a tedious manual process, as each 2D frame requires expert delineation of fine anatomical structures across the cardiac cycle. Most publicly available datasets, including EchoNet-Dynamic, provide labels only for end-diastole (ED) and end-systole (ES), the clinically most relevant phases. Extending to cross-dataset and phase-agnostic protocols, where the network must also infer cardiac phase, is a key direction for future work, and we will highlight this more clearly in the conclusion.
>
>
>
>
> We thank the reviewer again for recognizing the clarity and effort invested in this work and for recommending acceptance.

---

### Meta-Review · Area_Chair_9zck · 2025-11-02

**Recommendation:** Accept (Oral)
**Confidence:** 4

**Metareview:**

## Summary

This work explores the usefulness of video foundation models in echocardiography applications. The authors explore if the foundation models can be used for spatio-temporal landmark detection under different fine-tuning strategies: fully frozen, selective encoder unfreezing combined with graph convolutional network (GCN) and fully trainable. The experiments are comprehensive, and yield useful insights into the application of foundation models for medical imaging applications.

## Reviewer comments and rebuttal

All reviewers were positive of the work in their initial reviews. While they identified the lack of novel methodological contributions, all of them recognized the value in a well-done experimental validation study on an interesting and timely topic. Several minor concerns were raised by the reviewers which have been largely addressed by the reviewers.

Overall, the work adds useful empirical evidence to the usefulness of video foundation models for the specific task of spatio-temporal landmark detection. Using appropriate fine-tuning of few layers can yield state-of-the-art performance on a challenging task. These points are convincingly made in a well-conducted study and nicely written paper.

---

### Decision · Program_Chairs · 2025-11-05

**Decision:**

Accept (Oral)

**Comment:**

We recommend an oral and a poster presentation given the AC and reviewers recommendations.